# Virus-Host Dynamics in Archaeal Groundwater Biofilms and the Associated Bacterial Community Composition

**DOI:** 10.3390/v15040910

**Published:** 2023-03-31

**Authors:** Victoria Turzynski, Lea Griesdorn, Cristina Moraru, André R. Soares, Sophie A. Simon, Tom L. Stach, Janina Rahlff, Sarah P. Esser, Alexander J. Probst

**Affiliations:** 1Environmental Microbiology and Biotechnology (EMB), Department of Chemistry, Group for Aquatic Microbial Ecology, University of Duisburg-Essen, Universitätsstraße 5, 45141 Essen, Germany; 2Environmental Metagenomics, Research Center One Health Ruhr, University Alliance Ruhr, Universitätsstraße 150, 44780 Bochum, Germany; 3Faculty of Chemistry, University of Duisburg-Essen, Universitätsstraße 5, 45141 Essen, Germany; 4Institute for Chemistry and Biology of the Marine Environment (ICBM), Carl-von-Ossietzky-University Oldenburg, Carl-von-Ossietzky-Straße 9-11, 26111 Oldenburg, Germany; 5Centre of Water and Environmental Research (ZWU), University of Duisburg-Essen, Universitätsstraße 5, 45141 Essen, Germany; 6Centre for Medical Biotechnology (ZMB), University of Duisburg-Essen, Universitätsstraße 5, 45141 Essen, Germany

**Keywords:** deep biosphere, subsurface viruses, Altiarchaeota, fluorescence in situ hybridization, virusFISH, direct-geneFISH, microbial heterogeneity

## Abstract

Spatial and temporal distribution of lytic viruses in deep groundwater remains unexplored so far. Here, we tackle this gap of knowledge by studying viral infections of Altivir_1_MSI in biofilms dominated by the uncultivated host *Candidatus* Altiarchaeum hamiconexum sampled from deep anoxic groundwater over a period of four years. Using virus-targeted direct-geneFISH (virusFISH) whose detection efficiency for individual viral particles was 15%, we show a significant and steady increase of virus infections from 2019 to 2022. Based on fluorescence micrographs of individual biofilm flocks, we determined different stages of viral infections in biofilms for single sampling events, demonstrating the progression of infection of biofilms in deep groundwater. Biofilms associated with many host cells undergoing lysis showed a substantial accumulation of filamentous microbes around infected cells probably feeding off host cell debris. Using 16S rRNA gene sequencing across ten individual biofilm flocks from one sampling event, we determined that the associated bacterial community remains relatively constant and was dominated by sulfate-reducing members affiliated with *Desulfobacterota*. Given the stability of the virus-host interaction in these deep groundwater samples, we postulate that the uncultivated virus-host system described herein represents a suitable model system for studying deep biosphere virus-host interactions in future research endeavors.

## 1. Introduction

Microbes drive the biochemical cycling of nutrients and control food-web trophic interactions on Earth. The smallest biological entities, i.e., viruses, manipulate microorganism-driven biogeochemical processes by impacting host metabolism, host evolution, and microbial community composition [1]. By killing their hosts, viruses cause a transformation of microbial biomass into particulate organic matter (POM) and dissolved-organic matter (DOM) [2]. This “viral shunt” can mediate a shuttle of organic carbon from autotrophic to heterotrophic microbial communities for stimulating their growth [1] and generating a so-called “microbial loop” [3]. However, not all viral lifestyles are involved in host lysis but can have different effects on the host ecology [4]. While lytic viral infections are characterized by the rapid production of new virions followed by lysis of the host cells, lysogenic viruses integrate their genome into the host chromosome and proliferate via cell division of the host. Thus, lysogeny has been suggested as a survival strategy for viruses living at low host density and/or at low nutrient content [3,4,5]. Lysogeny is found to be common in, e.g., seawater, extreme environments, sediments, or hydrothermal vents [3,5]. As such, it has been proposed that lysogeny is the prevalent lifestyle of viruses in the deep (continental) biosphere [5], because viruses are challenged with finding a host in the deep subsurface, where microbial biomass is generally low [6]. Based on metagenomic and virus-targeted direct-geneFISH (virusFISH) recent investigations as in Holmfeldt et al. (2021) and Rahlff et al. (2021) [7,8] demonstrated that lytic viruses can be abundant in the deep biosphere and even target main primary producers following the “kill-the-winner” model [9]. For example, a drastic increase of virus-host ratios with depth was determined for marine sediment [10] and deep granitic groundwater [11]. In the aforementioned study, viral abundance was correlated with bacterial abundance in a ratio of 10:1 in samples collected from 69 to 450 m depth (10^5^–10^7^ virus-like particles mL^−1^ and 10^4^–10^6^ total number of prokaryotic cells mL^−1^, [11]).

Viruses of Archaea—as they have been reported in deep granitic groundwater [11]—are some of the least understood groups of viruses with unique morphologies compared to eukaryotic viruses or bacteriophages [12,13] (also reviewed in [14]). Knowledge on Altiarchaeota and their viruses has mainly been gained from the uncultivated genus *Candidatus* Altiarchaeum with the best studied representative *Ca.* Altiarchaeum hamiconexum [15], which is a frequent target of recently described lytic archaeal viruses in the deep subsurface [8]. Due to their worldwide distribution and high abundance as main primary producers in the deep subsurface (carbon fixation via a modified reductive acetyl-CoA pathway [16]) *Ca.* Altiarchaea have been heavily analyzed regarding their ecophysiology [16,17,18,19,20,21,22,23]. They often form nearly pure biofilms in the subsurface (>95% of the cells) [18] or streamers with a string-of-pearls-like morphology when associated with sulfur oxidizing bacteria in surface streams [21,22] and thus reach high abundances in their ecosystems constituting up to 70% of the total microbial community [8].

While the biology of viral attacks in biofilms is generally rather complex [24], biofilms also have the potential to be a hotspot for viral activity due to their high cell density [9]. However, the extracellular polymeric substances (EPS) of biofilms can also act as barriers against viral infection [25]. Furthermore, cell surface appendages as shown for the amyloid fiber network of *E. coli* can prevent viral infections with its lytic phage T7 [26]. At the same time, phages already trapped in the biofilm matrix may remain active and can also eliminate newly arriving prokaryotic cells [27]. It has also been suggested that viruses may enhance biofilm formation through the induction of polysaccharide production [28,29]. These examples illustrate the complexity of virus-host interactions in biofilms, yet little is known about how viruses affect biofilms and community structures in the deep biosphere. Biofilms with a high cell density could increase virus-host contacts enabling an easy spread of viral infections in the deep subsurface [5], however, actual evidence to verify or falsify this hypothesis is still missing.

For the present study, we used naturally grown biofilms of the uncultivated host *Ca.* A. hamiconexum that can be accessed through the Muehlbacher Schwefelquelle (MSI, near Regensburg, Germany) to answer the question of how virus-host ratios change over time and across individual biofilms. We took samples for four consecutive years from 2019 to 2022 (once per year) and applied virusFISH to biofilms dominated by the uncultivated virus Altivir_1_MSI and its host. Individual biofilm flocks from 2022 were analyzed using qPCR designed for the detection of *Ca.* A. hamiconexum, Altivir_1_MSI, and bacteria and archaea in general (excluding the host). While virus-host ratios showed a constant increase over the years, we generally observed strong heterogeneity regarding the infections in biofilms. We consequently propose a temporal succession from little to no infections in biofilms to high virus-host ratios that can be associated with the enrichment of filamentous microbes during cell lysis. Microbiome analyses based on full-length 16S ribosomal RNA (rRNA) gene analyses of individual flocks revealed a relatively constant community composition of the associated bacteriome in the biofilms.

## 2. Materials and Methods

### 2.1. Sampling Procedure and DNA Extraction

Biofilm flocks from the deep subsurface were collected from the cold (~10 °C), sulfidic spring (drilled to a depth of 36.5 m), Muehlbacher Schwefelquelle (Regensburg, Germany, 48° 59.142 N, 012° 07.636 E), as described previously [30]. We used Schott flasks with two openings (fused by the university’s glass blowing workshop) and inserted polyethylene nets for collecting enough biofilm flocks (~500 flocks per sampling event). The flask was placed on a funnel to make as much spring water flow through the nets as possible. This biofilm trapping system has the advantage that the biofilms directly stick to the nets due to the flow rate of the spring with ~5.50 m^3^ h^−1^ and due to their *hami* that represent cell surface appendage with nano-grappling hooks [16]. Each of the biofilm trapping systems were incubated for one day as deep as possible (~1 m) in the borehole. Further information on the environmental parameters of the sulfidic spring is described elsewhere and proved to be interannually constant [21,30].

For virusFISH, biofilms were collected in January 2019, August 2020, May 2021, and February 2022 (see Appendix A). Biofilm samples for DNA extractions and quantitative polymerase chain reaction (qPCR) experiments were taken in February 2022.

Genomic DNA was extracted from biofilm samples using the RNeasy^®^ PowerBiofilm Kit (Qiagen GmbH, Hilden, Germany) following the manufacturer’s instruction and a DNA-conform workflow. For accurate quantification of genomic DNA, a Qubit high-sensitivity DNA assay kit and a Qubit Fluorometer (Qubit 4, both Thermo Fisher Scientific, Waltham, MA, USA) was used, and the genomic DNA was stored at −20 °C until further use.

### 2.2. VirusFISH for Enumerating Viral-Host Ratios

VirusFISH and imaging were performed on 18 Altiarchaeota biofilm flocks for each of the four sampling events (*n* total = 72 biofilm flocks; raw data are listed in Appendix A) following the protocol of Rahlff et al. 2021 [8]. For enumerating 72 *Ca.* A. hamiconexum biofilms, 68 biofilms were treated with the Altivir_1_MSI probe (*n* = 68) and four biofilms were treated with a *Metallosphaera* sp. virus probe (*n* = 4). Shortly, the biofilms were hybridized with Atto 488 labelled 16S rRNA probes, Alexa 594 labelled virus probes, and counterstained with 4′,6-diamidin-2-phenylindole (DAPI, 4 µg mL^−1^, Thermo Fisher Scientific, Waltham, MA, USA). Of the 18 biofilms from each year, 17 were treated with the Altivir_1_MSI probe (*n* = 17; see Supplementary Information of Rahlff et al. 2021 [8]) and one served as negative control (*Metallosphaera* sp. virus probe [8,31]; Appendix A). Imaging was performed with an Axio Imager M2m epifluorescence microscope (X-Cite XYLIS Broad Spectrum LED Illumination System, Excelitas, ON, Canada) equipped with an Axio Cam MRm and a Zen 3.4 Pro software (version 3.4.91.00000) (Carl Zeiss Microscopy GmbH, Jena, Germany). The visualization was performed by using the 110×/1.3 oil objective EC-Plan NEOFLUAR (Carl Zeiss Microscopy GmbH) and three different filter sets from Carl Zeiss: #49 DAPI for detecting DNA, #64 HE mPlum for the detection of signals of probes targeting Alitvir_1_MSI, and #09 for visualizing 16S rRNA signals of *Ca.* A. hamiconexum. We calculated virus-host ratios by summarizing virus counts across all three defined infection stages and compared the value to the number of host cells in the specimen. We use this as a proxy for the infection frequency in a sample.

### 2.3. Determining the Detection Efficiencies of Direct-GeneFISH and VirusFISH

Three different probe sets targeting the pseudo-genome *Ca.* A. hamiconexum (NCBI acc. no. JAGTWS000000000.1), as assembled from the MSI [8], were designed for the experiments estimating the detection efficiency of direct-geneFISH [32]. Each probe set contained eleven dsDNA polynucleotides, having 300 bps in length.

In preparation for probe design, four metagenomic datasets from 2012 and 2018 [8,16] were mapped and ran in sensitive mode using Bowtie2 (v. 2.3.5.1) [33] to the reference genome of the host *Ca.* A. hamiconexum and its virus Altivir_1_MSI, respectively. Then, for each metagenomic sample and each host/virus genome inStrain (v. 1.5.3) [34] was used to (i) detect single nucleotide polymorphisms (SNPs), counting all positions that were classified as ‘divergent sites’; and (ii) calculate the per base coverage. For calculating the coverage, we removed (i) all 0 coverage regions during the pseudocontig creation by joining scaffolds with 1000 Ns as insert regions in-between: and (ii) all genome positions corresponding to an N.

Then, all genomic regions with high SNP counts (more than 5 SNPs in 300 base window, or more than 10 SNPs in a 30-base window) and/or low coverage (lower than the median coverage for the respective metagenome) were removed. Only regions that were found in all metagenomes were kept. For the remaining genomic regions, polynucleotides of 300 bases (N free) were generated. Only those polynucleotides with a G+C base content between 30% and 40% were kept, similar to the G+C base content of the polynucleotides used to target Altivir_1_MSI [8]. Their melting profiles were predicted using the DECIPHER R package [35]. To further aid the probe selection, we plotted the remaining polynucleotides along the length of the pseudocontig, together with their corresponding SNP counts (number of SNPs per polynucleotide) and coverage, for each metagenome. The plots were inspected visually and the polynucleotides in the three probe sets (Appendix A) were chosen using the following criteria: (i) localization on the same scaffold and within a 10,000 bases region, to ensure spatial proximity similar to that of the Altivir_1_MSI probes; (ii) per polynucleotide SNP counts similar to that of the Altivir_1_MSI probes, for which the SNP counts ranged of between 0 and 2.7 (see the plot of the Appendix A); and (iii) similar melting profiles (Appendix A).

The polynucleotides were chemically synthesized by IDT (Integrated DNA Technologies, CA, USA) as gBlocks^®^ Gene Fragments. All eleven polynucleotides from each probe set were mixed in equimolar ratios and then labelled as previously described [8], using the ULYSIS^™^ Alexa Fluor^™^ 594 nucleic acid labeling kit (Thermo Fisher Scientific, Waltham, MA, USA).

For targeting the 16S rRNA of *Ca.* A. hamiconexum a specific SM1- Euryarchaeon- probe “SMARCH714” (5′-GCCTTCGCCCAGATGGTC-3′, [36]) was used. As a negative control for the experiment, also *E. coli* was used for applying the different amounts of probe sets and their combinations (Appendix A). Five biofilm flocks for each of the three probe sets (probe set 1, 2, and 3) and each probe set combination (1 + 2, 2 + 3, 3 + 1, 1 + 2 + 3) were used (*n* = 35 in total). For details, please see Appendix A.

### 2.4. Quantitative Real-Time PCR (qPCR) Targeting Altivir_1_MSI and Archaeal as Well as Bacterial 16S rRNA Gene Sequences

A primer set for targeting the previously identified [8] viral genome “Altivir_1_MSI” (GenBank accession number #MW522970) was designed with Primer3 [37] resulting in Altivir_1_MSI_F (5′-CGATTACACTCACCGGCTTG-3′) and Altivir_1_MSI_R (5′-CGCTCCAACCACGAATGATT-3′) (Appendix A). The new primer set was evaluated against NCBI’s nr and available metagenomes of archaeal biofilm samples from the respective site [8,16] using blastn [38]. Archaeal 16S rRNA genes were targeted with primer set 345aF (5′-CGGGGYGCASCAGGCGCGAA-3′ [39]) and 517uR (5′-GWATTACCGCGGCKGCTG-3′ [40]) and archaea- and bacteria-directed 16S rRNA genes with 515F (5′-GTGYCAGCMGCCGCGGTAA-3′ [41]) and 806R (5′-GGACTACNVGGGTWTCTAAT-3′ [42]), which do not detect *Ca.* Altiarchaeum. qPCR standards were generated by amplifying the respective product from DNA from biofilms flocks, followed by cloning into *Escherichia coli* (TOPO^®^ Cloning Kit, Thermo Fisher Scientific, Waltham, MA, USA) and purifying the respective vector. Inserts of the vectors were confirmed via Sanger sequencing (Eurofins Genomics, Ebersberg, Germany).

Bacterial, archaeal, and Altivir_1_MSI abundances were estimated by qPCR in ten individual MSI biofilm flocks (42.6 to 126 ng of DNA per flock) collected in February 2022. DEPC-treated water was used as a template for negative controls. Dilution series of the respective vectors were used as positive controls (see above). Reactions were performed in triplicates for all samples (here MSI biofilm flocks) and in duplicates or triplicates for the respective standards (10^−1^–10^−8^ or 10^1^–10^9^ copies µL^−1^). The R^2^ values of the standard curves ranged from 0.96 to 0.99 (see Appendix A).

All qPCR reactions (20 µL) were performed in MIC tubes (Biozym Scientific GmbH, Hessisch Oldendorf, Germany) containing 18 µL master mix (2× qPCRBIO SyGreen Mix, PCR Biosystems Ltd., London, UK), 0.4 µM of the respective forward and reverse primer, 1 µL bovine serum albumin (BSA) per reaction (Simplebiotech GmbH, Leipzig, Germany), DEPC -treated water (Biozym Scientific GmbH, Hessisch Oldendorf, Germany) and 2 µL of DNA template. The thermal cycling steps were carried out by using a MIC qPCR cycler (Bio Molecular Systems, Queensland, Australia). For the primers targeting archaea and Altivir_1_MSI, they consisted of 95 °C for 2 min and 40 cycles at 95 °C for 30 s, 60 °C for 30 s, and 72 °C for 30 s. The qPCR steps for the archaea- and bacteria-directed 16S rRNA gene primers were set as follows: 95 °C for 10 min and 35 cycles at 95 °C for 10 s, 50 °C for 30 s, and 72 °C for 30 s.

### 2.5. Statistical Analysis

A Kruskal-Wallis test was used to find significant differences among the different data sets obtained by qPCR and virusFISH and was performed in R (version 4.1.2) [43]. If significant differences (*p* < 0.05) were observed, the post-hoc Dunn’s test was used (Appendix A).

### 2.6. Full Length 16S rRNA Gene Sequencing from DNA of Individual MSI Biofilm Flocks by Using Nanopore Sequencing

For Nanopore sequencing of bacterial 16S rRNA gene amplicons, we used the forward primer (5′-ATCGCCTACCGTGAC-barcode-AGAGTTTGATCMTGGCTCAG-3′) and the reverse primer (5′-ATCGCCTACCGTGAC-barcode-CGGTTACCTTGTTACGACTT-3′) from the 16S Barcoding Kit (1–24 Kit SQK-16S024, Oxford Nanopore Technologies (ONT), Oxford, UK) with some modifications in the protocol. Full-length 16S rRNA gene PCR was carried out in a total volume of 50 µL containing 15 µL DNA template, 5 U µL^−1^ Taq DNA polymerase (Takara, CA, USA), 1× of PCR buffer (10×, Takara, CA, USA), 10 µL of barcoded primer set forward/reverse from ONT, 200 µM deoxynucleotide triphosphates (dNTPs, Takara, CA, USA), 1 µg µL^−1^ BSA (Simplebiotech GmbH, Leipzig, Germany), 1% (*v*/*v*) dimethyl sulfoxide (DMSO, Carl Roth GmbH + Co. KG, Karlsruhe, Germany). Thermal cycling was carried out with an initial denaturation step (1) at 95 °C for 10 min, (2) denaturation at 95 °C for 30 s, (3) annealing at 54 °C for 30 s, (4) extension at 72 °C for 2 min (2–24 cycles), followed by an extension at 72 °C for 10 min. The resulting PCR products were purified using Agencourt AMPure^®^ XP beads (Beckman Coulter, IN, USA). The incubation with the magnetic beads was extended from 5 to 10 min.

The PCR product concentration in ng µL^−1^ per barcode, with barcode sequences listed in Appendix A. The 90.67 fmol of amplicons were loaded on an R9.4.1 FlowCell (ONT). Sequencing was performed for 48 h on a MinION1 kB sequencing device (ONT). Base calling and demultiplexing were performed using Guppy (v. 6.0.7, super high accuracy model). Basic sequencing statistics were collected using NanoPlot (v. 1.32.1) [44].

Demultiplexed reads were classified via a custom script by mapping to the SILVA 138 SSU (https://www.arb-silva.de/documentation/release-138/) database [45] (accessed on 13 January 2023) via minimap2 [46] allowing for up to ten mismatches. Reads mapping to the database were then clustered, and an OTU table was created with numbers of reads mapped to each reference sequence. The final OTU table contained SILVA 138 taxonomy for each reference sequence and the numbers of reads mapped to the reference sequence across all samples. In RStudio, OTUs assigned to “Chloroplast”, “Mitochondria” or “Eukarya” were removed manually and relative abundances were calculated. Finally, after summarizing total read numbers at genus level per sample (tidyverse [47]), ggplot2 [48] was used to generate heatmaps. 

## 3. Results

### 3.1. VirusFISH Reveals an Increase in Viral Infections of Ca. Altiarchaeum hamiconexum Cells in the MSI over Four Years

We compared the virus-host ratio of Altivir_1_MSI and *Ca.* A. hamiconexum in samples across four consecutive years (2019–2022) using a virusFISH protocol [8], in which we detected simultaneously the Altivir_1_MSI virus by using a set of eleven dsDNA polynucleotide probes, and *Ca.* A. hamiconexum, its host by using rRNA targeted oligonucleotides. To analyze spatial heterogeneity, i.e., different infection rates in individual biofilm flocks, we analyzed 18 biofilm flocks per sampling event. This resulted in the analysis of 55,827 individual cells, of which 2854 were infected (Figure 1A). We found the same three main infection categories as previously described [8]: (i) initial infections, represented by small, dot-like signals and including the viral adsorption, genome injection and early replication phases; (ii) advanced infections, displaying the so-called “halo” signals and including the advanced genome replication stages; and, (iii) lysing infections, recognizable from the virion release around the cells. While the percentage of advanced infections decreased over the years from 76.5 to 54.4%, initial infections (8.6–13.3%) and lysing infections (14.9–32.3%) constantly increased, except for the year 2021 (Figure 1A). In general, the virus-host ratio increased from 0.12 to 0.28 throughout the years 2019 to 2022 (median; Figure 1B, Appendix A). These absolute ratios agreed well with the virus-host ratios previously found in metagenomes of biofilms from 2012 and 2018 [8]. While the ratio determined for the 2018 metagenome (0.0033) aligned well with the increasing trend in virusFISH from 2019 to 2022, the ratio of 0.312 from the 2012 metagenome was extraordinarily high, however, still in the range of the virusFISH-based ratios observed across the years. In addition, we also calculated the virus-host ratio for ten individual biofilm flocks sampled in 2022 for analyzing the distribution of Altivir_1_MSI, *Ca.* A. hamiconexum, and the bacterial community composition (for more details please see Section 3.4). Here, the virus-host ratio ranged from 0.001 to 0.492, similar to the ratio obtained for the 17 biofilm flocks (Figure 1B). In Figure 1B, significant differences between populations are marked with an asterisk showing a *p*-value ≤ 0.01.

### 3.2. Determining the Detection Efficiency of VirusFISH via Host-Directed Direct-GeneFISH

We observed striking differences in the abundance of the three infection categories across all years, with the abundance of the initial infection stage being the lowest (8.6–13.3%; Figure 1A). The number of viral genomes per cell varies during the virus reproduction cycle. During the initial stage of infection, there can be as little as one viral genome copy per host cell. As the detection of single copy targets in virusFISH is at the limit of sensitivity, potentially resulting in decreased detection efficiencies, we investigated here whether the low numbers of initial infections stem from a low detection efficiency. For this, we designed a direct-geneFISH protocol (on which the virusFISH is based) that compares the detection efficiency of *Ca*. A. hamiconexum cells by genome-targeted polynucleotide probes with the detection of *Ca.* A. hamiconexum by 16S rRNA probes. The latter is known to have a 100% detection efficiency, based on previous publications [17,18,30]. Three different *Ca.* A. hamiconexum probe sets were designed, each having eleven polynucleotides and other similar properties (G+C base content, polynucleotide length, etc.—see Section 2 “Materials and methods”) with the probe mix targeting Altivir_1_MSI. To avoid targeting individual strains of *Ca.* A. hamiconexum, which are known to exist in MSI [16], the probe design was performed on genomic regions which had a coverage equal to or larger than the median coverage in four metagenomes [8].

We first applied each probe set targeting the host genome individually and retrieved a detection efficiency of 15.0–16.1% compared to 16S rRNA geneFISH (see Figure 2). Hybridizing with combinations of two or three probe sets, to obtain probe mixtures of 22 and 33 polynucleotides, showed a linear increase of the detection efficiency. The highest efficiency was obtained for the 33-polynucleotide mixture, with an average detection efficiency of 43.5% (for the calculation see Appendix A). Transferring these results to virusFISH, where we target a single Altivir_1_MSI genome by using eleven probes, means that more than three viruses in close vicinity are needed to reach a detection efficiency greater than 50%. In other words, at least seven viruses in close vicinity are necessary to achieve a detection efficiency of 100% when extrapolating these findings. Therefore, it is likely that our virusFISH results have underestimated the number of initial infections (category 1) by a factor of ~6.7, but not the number of advanced and lysing infections, which are expected to have more than ten viral genome copies per cell.

### 3.3. Filamentous Microorganisms Are Enriched in Areas of Vast Viral Lysis Suggesting a Development of Ca. A. hamiconexum Biofilms over Time

Using virusFISH, we observed different degrees of infection with Altivir_1_MSI of the *Ca.* A hamiconexum biofilms (*n* = 68). For some biofilm flocks (two out of 68), we observed a high accumulation of filamentous microorganisms in areas where *Ca.* A. hamiconexum showed heavy infections and viral lysis (Figure 3D). Although a similar accumulation of bacteria was generally observed in biofilm flocks (Figure 3A–C), their abundance was low [8]. In concert with results published previously [8], these results demonstrate that *Ca.* A. hamiconexum biofilms from MSI are homogeneous in terms of the associated bacterial community composition. We further suggest that the biofilms undergo a temporal development dependent on viral infections, with the main stages depicted in the individual panels in Figure 3. Initially, the biofilm shows no to very few viral infections (based on all infection categories, 17 out of 68 imaged flocks across 2019–2022, Figure 3A). Then, the infection frequency increases (47 out of 68, Figure 3B), until the vast majority of cells are infected (two out of 68 biofilm flocks, Figure 3C). Finally, many *Ca.* A. hamiconexum cells lyse and filamentous microbes enrich along with the cell debris (two out of 68 biofilm flocks, Figure 3D). Out of 68 biofilms, 64 individual biofilms (94%) were infected, and four biofilms (6%) had no detectable infections (for raw data please see Appendix A). Filamentous microbes that often appeared along with cell lysis are likely bacteria, as indicated by the previous identification as such of organisms with similar morphology [30], and by results based on 16S rRNA gene amplicon analysis (see below).

### 3.4. Biofilm Flocks with Different Virus-Host Ratios Are Associated with a Constant Bacteriome

To investigate bacteria associated with biofilms with different infection frequencies of the *Ca.* Altiarchaea cells, we sampled ten individual flocks from 2022 and investigated the virus-host ratio, bacterial abundance, and bacterial community composition using near full-length 16S rRNA gene amplicon sequencing (Figure 4A,B). Our qPCR analysis (Figure 4A) displayed already a constant distribution of archaea (mainly *Ca.* A. hamiconexum) ranging from 7.07 × 10^8^ to 2.52 × 10^9^ copies per µL, Altivir_1_MSI from 1.26 × 10^6^ to 4.52 × 10^8^ copies per µL, and bacteria from 5.93 × 10^5^ to 2.66 × 10^6^ copies per µL in the ten individual biofilm flocks. The virus-host ratio varied from 0.001 to 0.492 based on specific qPCR assays. The ratio of *Ca.* A. hamiconexum to bacteria varied from 569.524 to 1240.105 (raw data in Appendix A). Apart from *Ca.* A. hamiconexum, the community comprised only bacteria, and was dominated by organisms of the genus *Desulfocapsa*. However, there was no significant correlation between Altivir_1_MSI abundance and bacterial abundance, the community composition, or the top three most abundant organisms (Appendix A). *Proteobacteria* (class: *Desulfobacterota*) was the most predominant phylum, accounting for up to ~76% of the total relative abundance. *Desulfocapsa* was present in all ten biofilm samples with 20–75.7%. Six biofilms showed also hits for another, potentially sulfate-reducing bacterial clade belonging to *Desulfovibrio*, with a relative abundance between 5.4 and 14.4%. As a third group of organisms associated with sulfate-reducing bacteria, we detected a member of the *Desulfobacterium catecholicum group*, with 12.2% of the total relative abundance within one single biofilm. Other bacteria, e.g., *Spirochaetota*, were also present in relatively low abundance (5.6%) in one single biofilm flock besides some uncultivated/unclassified bacteria (rel. abundance between 5.0–32.7%). Next to many uncultivated bacteria, we also found hits for members of the phylum *Bacteroidota*, accounting for 11.3% of the relative abundance, and, e.g., *Lentimicrobium* (also a representative of the phylum *Bacteroidota*) with a relative abundance of 10.2 and 11.8% detected in two biofilm flocks, in which the abundance of Altivir_1_MSI was also high. We conclude that the bacteriome of these ten individual biofilm flocks displayed a stable community consisting of heterotrophic and potentially sulfate-reducing bacteria.

## 4. Discussion

The deep biosphere harbors by far the largest reservoir of organic carbon on Earth [49,50] and is estimated to contain 6 × 10^29^ prokaryotic cells [6], including members of not-yet cultivated bacteria and archaea. Despite the ecological and biogeochemical importance of prokaryotes in the deep biosphere, there is little information about the temporal succession and spatial distribution of their mortality due to viral attacks. In a previous study, we linked metagenomics and virusFISH to study one specific virus-host system from the deep subsurface, here *Ca.* A. hamiconexum and its virus Altivir_1_MSI [8]. Building upon this knowledge, we now investigated the stability of the virus-host system using virusFISH, over a period of four consecutive years. We have confirmed previous results on the different stages of viral infection on *Ca.* A. hamiconexum, but we also revealed a constant increase in virus-host ratios, i.e., potential infection frequency, over the years 2019 until 2022.

Comparing the virusFISH results with those from other methods, i.e., qPCR and metagenomics, revealed substantial differences, although results were in the same order of magnitude. These differences likely stem from the different sampling times (see Figure 1B) and also, from the specific biases that each method has, e.g., DNA extraction efficiency, primer binding efficiency, and labeling rate in fluorescence microscopy. Moreover, the size and the thickness of the biofilm can vary substantially, potentially leading to a reduction of virusFISH signals in thicker biofilm samples, due to the relatively strong autofluorescence of the EPS. However, we only used thin biofilm samples consisting of a few cell layers for our analyses and their heterogeneity revealed via virusFISH showed significant differences in infection frequencies, i.e., from no detectable infection to >94% of infections across different biofilms from different years. Consequently, the virus-host ratio determined via metagenomics—often based on sampling hundreds of biofilm flocks across several sampling campaigns [16]—might rather reflect an average ratio, while qPCR is the more suitable technique to resolve virus-host ratios across individual biofilm flocks. However, qPCR does not allow us to differentiate between the different infection stages within the biofilm. Since quantitative PCR methods and relative abundance measures of metagenomics usually correlate well for deep biosphere communities [22] and virusFISH helps to unravel the underlying infection stages, these approaches are complementary for determining the heterogeneity of such biofilms.

By investigating 68 individual biofilms flocks via virusFISH over four years, we identified two flocks that were heavily infected (nearly every cell with a virus signal) and another two with many lysis states. The latter two also revealed the accumulation of many filamentous microorganisms.

Moreover, *Ca.* A hamiconexum were previously found to be associated with filamentous bacteria identified as *Sulfuricurvum* sp., which potentially lives in a syntrophic relationship with *Ca.* A hamiconexum in oxygenated biofilms at the spring outflow [21]. While the sulfur oxidation of *Sulfuricurvum* sp. is usually tied to oxygen reduction, species of this genus have been reported to oxidize sulfur compounds with nitrate [51], which has been reported in the spring water [21]. Consequently, these filamentous structures around viral bursts could correspond to *Sulfuricurvum* sp., as sequences classified as such were also detected in our 16S rRNA gene analysis.

Furthermore, Lentimicrobia, which are usually strictly fermentative bacteria [52] but can also participate in sulfate reduction [53], were found in two biofilms. However, none of the abundant bacterial taxa that were detected in the 16S rRNA gene survey correlated with the viral abundance from qPCR (Appendix A). Nevertheless, we further suggest that some bacteria (e.g., the unclassified and/or other 5%) could also feed off cell remnants from lysed *Ca.* A. hamiconexum. These findings corroborate our previous hypothesis that viral lysis of primary producers likely jump-starts heterotrophic carbon cycling in the deep biosphere [8], and it may be a small fraction of the entire bacterial community that particularly benefits from virus-mediated nutrient release.

Having images of 68 biofilm flocks at hand, we developed a theory for the temporal succession of infection, starting with low to no viral signals, and ending up with many lysis states that involve filamentous microbes. We found that most infections were categorized as advanced infections, which is probably because this infection stage has the longest duration and is consequently more often captured than initial infections or lysing infections. Not only the detection efficiency underestimates initial infections, but also the lower chance of observing such stages likely affected our results. Due to the proximity of cells in biofilms, viruses might easily jump between hosts without necessarily starting immediate infections. The biofilm itself could limit viral dispersion, since a study on virus-host dynamics in a microbial mat found that the lowered mobility in a biofilm would rather support lysogeny as the predominant viral lifestyle [54], which is clearly not the case in the MSI ecosystem. Furthermore, Rahlff et al. 2021 described that *Ca.* A. hamiconexum uses adaptive immune defense in the form of clustered regularly interspaced short palindromic repeats (CRISPR)-Cas system to defend itself from Altivir_1_MSI, and hence we expected the observed oscillating infection frequencies being typical for a reasonably stable virus-host arms race [8]. Active CRISPR defense will also mean that many viruses in the adsorption stage (category 1), will finally not successfully replicate in the host. However, virus-host ratios clearly increased over the years as did the number of observed viruses undergoing adsorption. This suggests that the virus tries to counteract the host defense by heavy proliferation, facilitating chances for mutations that would allow circumventing the host’s armor. Increasing numbers of SNPs were already observed in Altivir_1_MSI between 2012 and 2018 [8].

The heterogeneity of infection frequencies in biofilms flocks of *Ca.* A hamiconexum further suggests that biofilms with a high cell density increase host cell contact, enabling a heavy spread of viruses after lysing their hosts. Consequently, these results support the hypothesis by Anderson et al. 2011, that this mode of viral dispersal and predation plays an important role in the deep biosphere [5]. Since previous studies about deep subsurface viruses do not provide information on ecosystem dynamics [7,55], the year-long stability of *Ca.* A hamiconexum and its virus Altivir_1_MSI described herein render their interaction, a perfect model system for studying virus-host interactions of aquatic samples from the deep biosphere.

## 5. Outlook

Despite the promising results of *Ca.* A. hamiconexum and its virus Altivir_1_MSI regarding their detection in situ, the virus-host ratio, and the infection frequency, many questions remain unanswered. First and foremost, the virion structure of Altivir_1_MSI has still not been identified and the current knowledge ends with bioinformatic analyses and virusFISH tagging. For linking the viral genome to the corresponding viral morphology, different possible approaches can be carried out in the future. One approach might be linking virusFISH and atomic force microscopy or scanning electron microscopy (SEM; [56]). Another promising approach could be Raman microspectroscopy for identification of infected *Ca.* A. hamiconexum cells [57] and its coupling to embedding and transmission electron microscopy (TEM). While these approaches are based on correlative microscopy, traditional immunogold labeling of viral proteins in MSI biofilms will eventually lead to the discovery of the actual virion structure visualized by TEM. With a specific method for detecting Altivir_1_MSI virions in fluorescence microscopy at hand, additional surveys regarding their occurrence and the accumulation of filamentous bacteria along with *Ca.* A. hamiconexum cell debris would bolster studying the viral ecology in this deep biosphere model ecosystem.

## Figures and Tables

**Figure 1 viruses-15-00910-f001:**
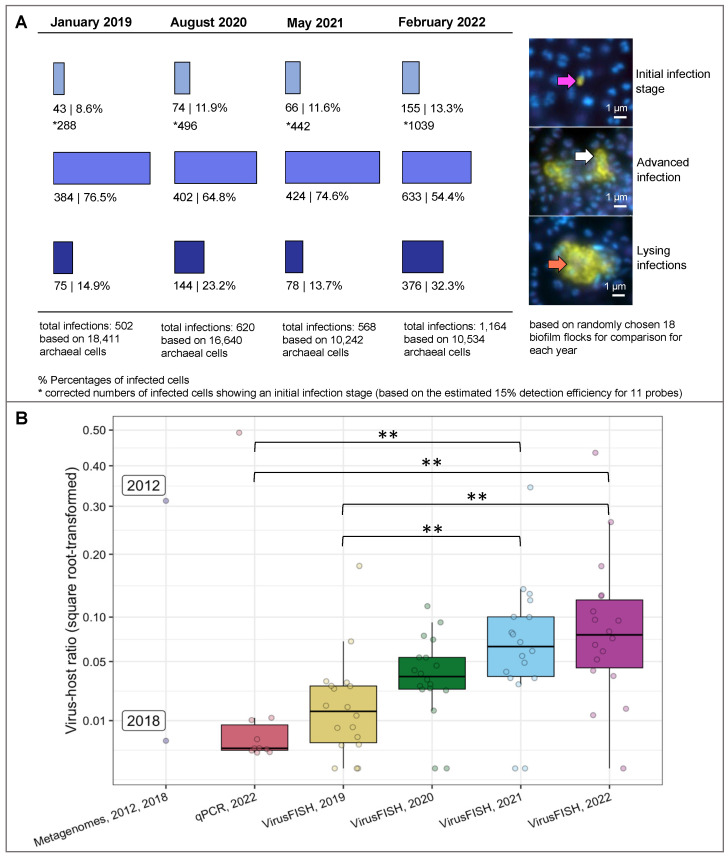
VirusFISH-based enumeration of infections of *Ca.* A. hamiconexum with Altivir_1_MSI across multiple years (**A**) and differences in virus-host ratios across techniques and years (**B**). (**A**) VirusFISH was performed by using a specific probe targeting Altivir_1_MSI within 18 altiarchaeal biofilms from each year sampled from the MSI [8]. The enumeration was conducted manually (data are listed in Appendix A). The biofilms were visualized by using three different fluorescent channels: DAPI (blue, archaeal cells), ATTO 488 (purple, SMArch714, 16S rRNA signal), and Alexa 594 (yellow, the probe mix specific for the Altivir_1_MSI genome). The three fluorescent channels were merged for visualization (individual images available on FigShare). Each of the merged micrographs shown here represent a different stage of viral infection. Purple arrow—virus attachment to the host’s cell surface. White arrows—advanced infections with “halo” signals. Orange arrows—cell burst, and release of free virions, according to Rahlff et al. 2021 [8]). Scale bars: 1 µm. The number of initial infections were corrected (indicated by an asterisk) by a factor of ~6.7 (100% detection efficiency/15% calculated detection efficiency of direct-geneFISH), but not the number of advanced and lysing infections, which are expected to have more than ten viral genome copies per cell. For details on infection frequency please see next paragraph in the main text. (**B**) The distribution of virus-host ratios across biofilm samples was determined using: (i) metagenomic read-mapping (data from Probst et al. 2013 [16] and Rahlff et al. 2021 [8]) on samples from 2012 and 2018; (ii) qPCR on ten individual biofilm flocks from 2022; and, (iii) the results obtained by virusFISH from 2019, 2020, 2021, and 2022 (data corresponds to panel (**A**)). The Kruskal-Wallis and post-hoc Dunn’s test were used to compare the virus-host ratios across qPCR and virusFISH datasets. Highly significant differences between populations are indicated with asterisks (*p* ≤ 0.01). For details, please see Appendix A. Different colors indicate different years or methods.

**Figure 2 viruses-15-00910-f002:**
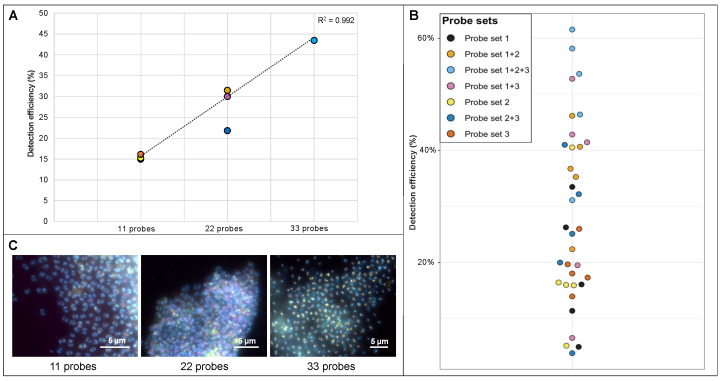
Determination of the detection efficiency of direct-geneFISH by using different probe sets targeting the genome of *Ca.* A. hamiconexum. (**A**) Three probe sets (probe set 1, 2 and 3 consisting of eleven probes each, Appendix A) were designed based on the *Ca.* A. hamiconexum genome (see methods for details). The sets were combined to create probe mixtures with 22 polynucleotides (probe set 1 + 2, 2 + 3 and 3 + 1) and a probe set with 33 polynucleotides in total (1 + 2 + 3). For each probe set (1, 2, and 3) and probe set combination (1 + 2, 2 + 3, 3 + 1, 1 + 2 + 3), five biofilm flocks were used (*n* = 35). The different amounts of polynucleotides in a mixture were positively correlated with the detection efficiency (*R*^2^ = 0.992, linear regression analysis). (**B**) Illustration of the detection efficiency in a bee swarm plot. The detection efficiency increases with increasing number of polynucleotides in a probe mixture (raw data can be found in Appendix A). (**C**) Visualization of the different detection efficiencies using fluorescence micrographs according to Figure 1. Biofilms were visualized by using three different fluorescent channels that were merged together: DAPI (blue, archaeal cells), ATTO 488 (purple, SMArch714, 16S rRNA signal), and Alexa 594 (yellow, probes targeting the *Ca.* A. hamiconexum genome). Scale bars: 5 µm. For unmerged imaging data see Appendix A. Raw image data available through FigShare. Scale bar 5 µm.

**Figure 3 viruses-15-00910-f003:**
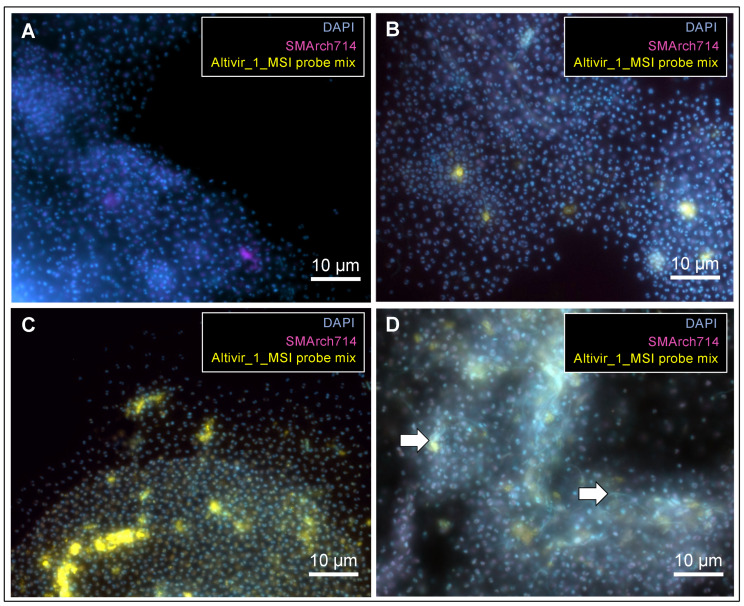
VirusFISH shows how infections of Altiarchaeota with Altivir_1_MSI in biofilms could progress over time. Here independent biofilm flocks with different infections frequencies are depicted. Infection frequencies are based on viral abundances derived from all infection categories (1–3). (**A**) Biofilms show no to very few viral infections (low infection frequency). (**B**) The infection frequency increases. (**C**) The vast majority of host cells are infected. (**D**) Lysis of *Ca.* A hamiconexum cells appears to promote the enrichment of filamentous microbes along with cell debris. White arrows indicate filamentous microbes. Micrographs were taken according to Figure 1 and raw data is available under FigShare. For unmerged imaging data see Appendix A. Scale bar: 10 µm.

**Figure 4 viruses-15-00910-f004:**
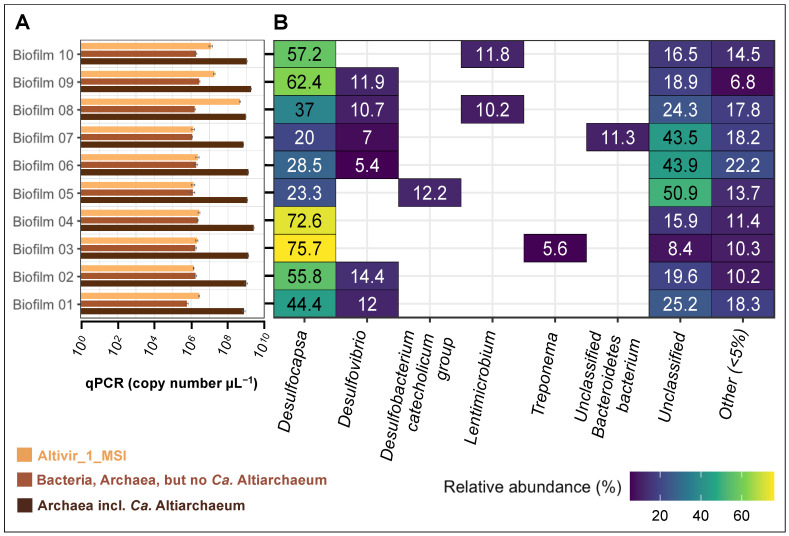
Quantification of microbes and bacterial community composition of ten individual biofilm flocks from MSI sampled in 2022. (**A**) qPCR data targeting archaea, bacteria, and Altivir_1_MSI of ten individual MSI biofilm flocks. For raw data please see Appendix A. (**B**) Community heatmap depicting the relative abundance (%) of the bacterial taxa at the genus level based on 16S rRNA gene information derived from individual MSI biofilm flocks.

## Data Availability

Microscopy and Nanopore data are available on Figshare (https://figshare.com/) (accessed on 23 March 2023) with the project title “Virus-host dynamics in archaeal groundwater biofilms and the associated bacterial community composition”.

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
