# Peer review of "Virus-Host Dynamics in Archaeal Groundwater Biofilms and the Associated Bacterial Community Composition"

_viruses, 2023, doi:10.3390/v15040910_

Round 1

Reviewer 1 Report

The manuscript presented very interesting experimental design and results over a relevant topic as lytic viruses in deep groundwater.

The work overall was easy to read and follow, methods are clearly explained as well as the results. Minor revision should be made to improve the understanding. Please, find the attached pdf with my comments and suggestions.

Reviewer 2 Report

Turzynski et al use virusFISH to study the spatial and temporal distribution of lytic viruses in groundwater. The manuscript is well written and will advance the field of microbial ecology’s knowledge on virus-microbe dynamics in deep subsurface microbial systems. I have no main issues with the manuscript since the authors use appropriate sampling and analytic techniques and do not overstate their findings. I only have two minor comments.

1.       Could the authors provide a more descriptive overview about their sampling procedure? Since the sampling procedure is integral to assessing, understanding and repeating their work a more detailed description should be provided. References are provided but one of the references refers to the other reference for more detail.

2.       I could not access the Supplementary Materials because I got an ‘Error 404 - File not found’ message anytime I tried. Could the authors check and fix this?
